# Applying Spinal Cord Organoids as a quantitative approach to study the mammalian Hedgehog pathway

**Markus Holzner** [ID]**, Anton Wutz** [ID]**\*, Giulio Di Minin** [ID]**\***

Department of Biology, Institute of Molecular Health Sciences, Swiss Federal Institute of Technology ETH Hönggerberg, Zurich, Switzerland

\* awutz@ethz.ch (AW); giulio.diminin@biol.ethz.ch (GDM)

## Abstract

The Hedgehog (HH) pathway is crucial for embryonic development, and adult homeostasis. Its dysregulation is implicated in multiple diseases. Existing cellular models used to study HH signal regulation in mammals do not fully recapitulate the complexity of the pathway. Here we show that Spinal Cord Organoids (SCOs) can be applied to quantitively study the activity of the HH pathway. During SCO formation, the specification of different categories of neural progenitors (NPC) depends on the intensity of the HH signal, mirroring the process that occurs during neural tube development. By assessing the number of NPCs within these distinct subgroups, we are able to categorize and quantify the activation level of the HH pathway. We validate this system by measuring the effects of mutating the HH receptor PTCH1 and the impact of HH agonists and antagonists on NPC specification. SCOs represent an accessible and reliable *in-vitro* tool to quantify HH signaling and investigate the contribution of genetic and chemical cues in the HH pathway regulation.

## Introduction

Hedgehog (HH) signaling is of great importance in embryonic development controlling morphogenesis, organogenesis, and the organization of the central nervous system [1]. Deregulation of the pathway is associated with congenital defects in the nervous, cardiovascular, or musculoskeletal systems [1]. Conversely, increased HH signaling contributes to cancer development as observed in basal cell carcinoma of the skin or medulloblastoma in the brain [2].

The HH pathway involves multiple layers of repressive interactions. The main transducer of HH signals is Smoothened (SMO), which belongs to the family of G-protein coupled receptors (GPCR). When the pathway is inactive, SMO is repressed by the HH receptor Patched1 (PTCH1) and restricted to intracellular compartments. Binding of one of the three mammalian Hedgehog ligands (Desert, Indian, or Sonic HH) to PTCH1 initiates the cascade by releasing SMO. SMO translocates to the plasma membrane, where it further progresses into the primary cilium. Here, SMO facilitates post-transcriptional modifications of the glioma-associated oncogene homolog (GLI) transcription factors which promote the transcription of HH

**Data Availability Statement:** All relevant data are within our paper.

**Funding:** GDM was supported by the ETH Zurich Postdoctoral Fellowship Program as well as the

Marie Curie Actions for People COFUND Program. This work was supported by grants from the Swiss National Science Foundation (SNF grants 31003A_152814/ 1 and 31003A_175643/1) to AW. The funders had no role in study design, data collection and analysis, decision to publish, or preparation of the manuscript.

**Competing interests:** The authors have declared that no competing interests exist.

target genes [3]. In the absence of HH signals, GLI proteins are subjected to degradation to their repressor forms (GliR). SMO activation prevents GLI processing at the primary cilia and allows the accumulation of its active form (GliA) [3].

Genetics has played a central role in the discovery and mechanistic understanding of HH signaling [4]. Many components of the HH cascade have been identified by genetic screens in *Drosophila*. Several of these factors have been studied subsequently in mammals suggesting the conservation of crucial players throughout evolution. However, divergence between *Drosophila* and vertebrates has been observed and requires systems for studying specific aspects of mammalian HH signaling [4]. In particular, primary cilia are crucial to vertebrate HH signaling but dispensable in *Drosophila* [4]. Additionally, HH signaling patterns have different anatomical features in invertebrates or vertebrates [4]. For example, the wing disc and neural tube are unique to *Drosophila* and vertebrates, respectively.

Studying the neural tube provides a valuable opportunity to investigate the effects and mechanisms of HH signaling during vertebrate development [5]. During neural tube development, a Sonic Hedgehog (SHH) gradient initiated at the notochord and later also promoted by the floor plate, defines the ventral domains of the neural tube. The intensity and duration of HH pathway activation leads to the expression of a different set of transcription factors, which can be considered as markers along the dorsal-ventral axis. SHH triggers a gradient of GLI activity in the neural tube that contributes to establishing the "GLI code" [6, 7]. This occurs by reducing GLI inhibitory effects and enhancing its function as an activator in a gradual manner. *Pax7*, *Pax6*, and *Irx3* expression occurs in the dorsal compartment and is antagonized by SHH signaling. In the ventral compartment, expression of *Nkx6.1*, *Olig2*, *Nkx2.2*, and *Foxa2* requires SHH signaling. While the absence of GliR is sufficient for *Nkx6.1* and *Olig2* transcription, the expression of the most ventral genes *Nkx2.2* and *Foxa2* requires additional GliA functions [6]. The classification and quantification of marker gene expression can be employed as a tool to infer HH pathway activity and effectively identify new genetic modulators of the HH pathway.

Analysis of markers in embryonic neural tubes is challenging. Firstly, for each investigated modulator, mutations would need to be established in mice. Preparing embryos and dissecting neural tubes requires effort, time, and funds. Secondly, genes that are crucial in early development, or neural tube specification cannot be analyzed if embryos do not survive until E9.5. This has led to the consideration of cell culture models for studying HH signaling. Current cell-based assays, including mouse embryonic fibroblast (MEFs, 3T3s), are widely used to mechanistically dissect the HH pathway by analyzing GLI-mediated transcriptional effects and ciliary localization of SMO. However, these systems are limited in understanding HH function in tissue patterning. Spinal Cord Organoids (SCO) are an *in vitro* system that recapitulate to a large extent the regulation of neural tube patterning and hence capture relevant aspects of HH signaling in a physiological context [8]. Importantly, the repressive function of GliR can be assessed. SCOs can be obtained from embryonic stem cells (ESCs) within six days. Although SCOs do not show a regional pattering, the specification of neural progenitors with dorsal and ventral identities is observed, similar to the neural tube.

In our recent work [9], we apply SCOs obtained using a method previously established by Wichterle and colleagues [8] to quantify HH signaling. We used this approach to define the role of the COP-1 receptor TMED2 in the HH signaling cascade. Here, we provide a detailed description of the procedure, including several optimizations, to generate homogenous SCOs and to monitor HH pathway activity by assessing specific marker gene expression for different positions along the dorsal-ventral axis.

## Material and methods

The protocol described in this peer-reviewed article is published on protocols.io, dx.doi.org/10.17504/protocols.io.kxygx3r6og8j/v2 and is included for printing as S1 File with this article.

## Expected results

Starting from 129-mouse ESCs, this protocol first generates neural embryonic bodies (nEBs) and later Spinal Cord Organoids (SCOs) (Fig 1A). ESCs are plated in AggreWell plates to form uniform nEBs, which are then transferred to 10cm dishes for SCO maturation (Fig 1B). Supplemented retinoic acid (RA) promotes neuralization and a caudal identity of neural progenitors (NPCs), and the subsequent addition of SHH triggers the maturation of ventral versus dorsal phenotypes of NPCs (Fig 1E). The extent of this ventral signal can be later used to measure the effect of gene mutations and compounds on the HH signaling cascade.

We characterized HH signaling in SCOs on two different levels; first, as the gene expression profile on the level of expressed mRNA, and second, as protein expression, which we detect by immunofluorescence (IF). Within our qPCR analysis, we evaluated the expression of transcription factors that are involved in the differentiation of the neuroectoderm and the specification of neural tube progenitors. *Sox1*, which is also expressed in the neural tube of E9.5 embryos, highlights the stemness and neural identity of the obtained SCOs [10]. Compared to ESCs, SCOs show a strong upregulation of *Sox1* mRNA (Fig 1F), indicating a successful differentiation into NPCs. We next evaluated the expression of the canonical HH target genes *Gli1* and *Ptch1*. Upon SHH stimuli, we observed a significant boost in *Gli1* and *Ptch1* expression (Fig 1C), indicating heightened GLI transcriptional activity. Activation of canonical HH signaling is also confirmed by the translocation of SMO to the primary cilium (PC) upon SHH stimulation (Fig 1D). The third group of genes we analyzed includes *Pax7* and *Pax6* and characterizes the dorsal fraction of the neural tube where *Pax7* is only expressed in the most dorsal regions [5]. Compared to ESCs, the SCOs display dorsal characteristics marked by high expression of both *Pax7* and *Pax6* (Fig 1F). As expected, upon HH stimuli, we observed a decrease of these markers concurring with the first steps of the HH pathway activation. Lastly, we measured the expression of the ventral markers *Olig2*, *Nkx2.2*, and *Foxa2*, where *Foxa2* defines the floorplate, the most ventral part of the neural tube [5]. These three markers are exclusively expressed in SHH-treated SCOs and thus confirm the ability of SCOs to recapitulate the full spectrum of progenitor fates during the dorsal-ventral patterning of the neural tube (Fig 1F).

In the next set of experiments, we quantified the HH response by analyzing the fraction of neural progenitors with either dorsal or ventral identity by staining through IF SCO sections for SOX1, PAX6, OLIG2, and NKX2.2 markers. As shown in Fig 2A, SCOs show a percentage of SOX1+ cells greater than 90%, essentially yielding a nearly homogenous population of neural progenitors. These NPCs show a mainly dorsal identity, indicated by the high proportion of PAX6+ cells. After SHH treatment, NPCs with the ventral markers OLIG2 and NKX2.2 are enriched (Fig 2B and 2C). The observation of a larger fraction of NPCs expressing OLIG2 than NKX2.2 recapitulates the *in vivo* situation where activation of NKX2.2 depends on higher SHH levels compared to OLIG2. Notably, the fraction of OLIG2+/NKX2.2+ NPCs is extremely low (Fig 2D), as the expression of NKX2.2 inhibits *Olig2* transcription. By treating SCOs with different concentrations of SHH, we were able to analyze varying grades of HH signaling intensity (Fig 2E–2G). The maximum fraction of NKX2.2 positive cells was achieved with the highest SHH concentration, while the top OLIG2+ NPC frequency was obtained at lower SHH concentrations.

To investigate the use of SCOs for characterizing HH pathway modulators, we generated SCOs from mouse *Ptch1*$^{KO}$ ESCs and analyzed their patterning along the dorsal-ventral axis.

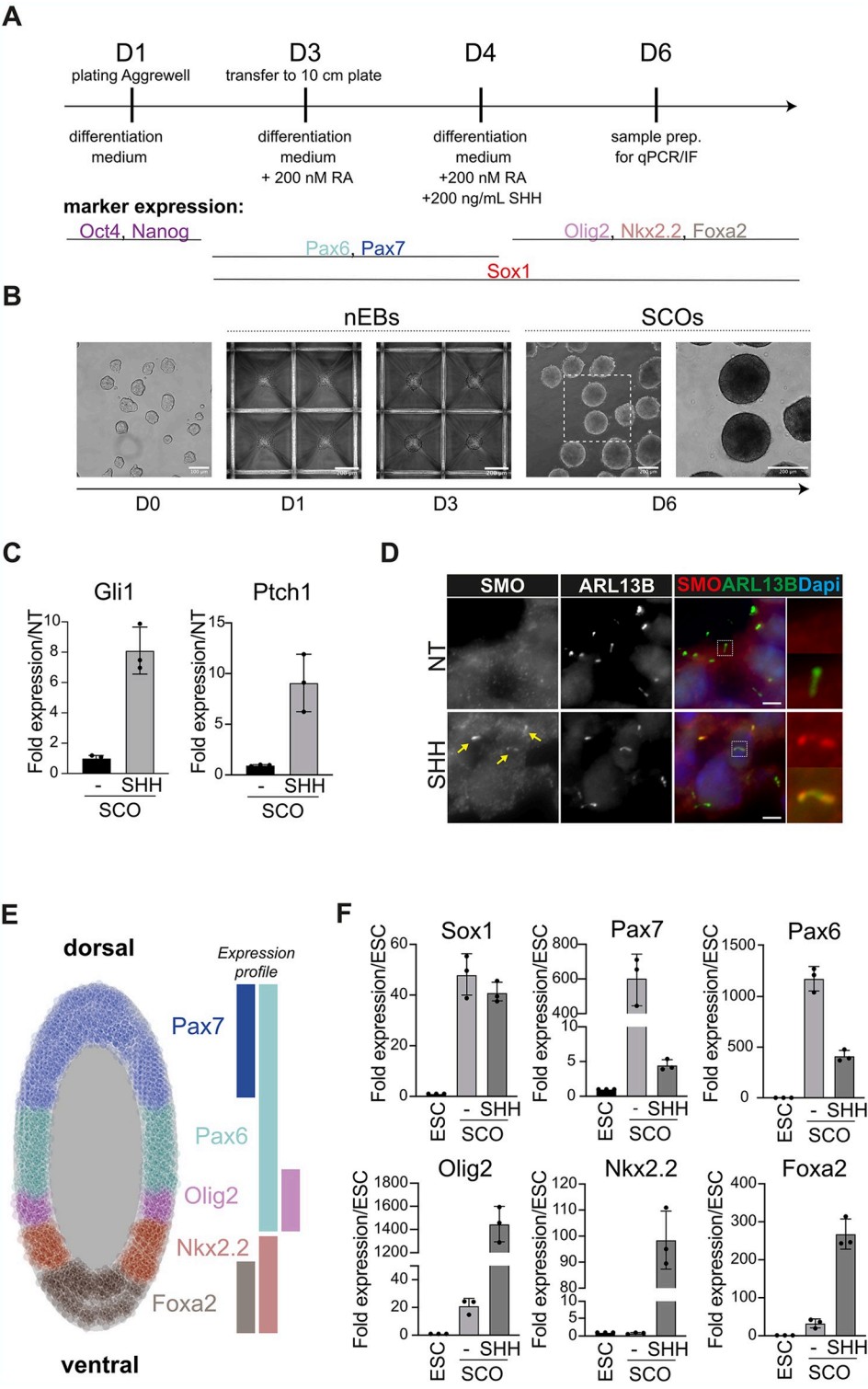

**Fig 1. Derivation of SCOs and analysis of their transcriptional profile. A)** Schematic overview of SCO differentiation process and expected neuronal marker expression during the differentiation. **B)** Brightfield microscopy of ESCs, nEBs and SCOs during D1, D3 and D6, respectively of differentiation. D1 ESCs grow as colonies on gelatine in 2i, Lif conditions, scale bar 100 μm; D1 and D3 nEBs grow in AggreWell Plate, first as single cells and then as nEBs, scale bar 200 μm; D6 SCOs grow in suspension, right frame shows zoom, scale bar in both frames 200 μm **C)** SHH treatment induces expression of HH targets in SCOs. Analysis by RT-qPCR of *Gli1* and *Ptch1* mRNA expression in

SCOs treated with and without SHH. Fold induction is relative to the untreated sample (mean ± SD). Each dot represents a biological replicate. **D)** Translocation of SMO to the PC in SCOs upon SHH stimulation. ARL13B is shown as a PC marker. SCOs were treated for 24 hours with and without SHH. Nuclei are labeled with Dapi. Scale bar 2,5 μm. **E)** Model mouse E9 neural tube with the expected marker pattering along the dorsal-ventral axis. **F)** Marker expression in SCOs at day 6 upon HH stimulation. Analysis by RT-qPCR of neuroectoderm (*Sox1*) markers with a dorsal (*Pax6* and *Pax7*) or ventral (*Olig2*, *Nkx2.2*, and *Foxa2*) identity. Fold induction is relative to the ESCs sample (mean +/- SD). *Eif4a2* expression was used for normalization. Each dot represents a biological replicate.

*Ptch1*[KO] mice die in utero at around E9.5 displaying severe deformations in cephalic regions, including defects in the neural tube closing [11]. Sections of these tubes display a strong ventral identity where NKX2.2 expression is also detected in dorsal domains. Here, SCOs derived from *Ptch1*[KO] ESCs show a high percentage of OLIG2 (64%) or NKX2.2 (18%) expressing NPCs (Fig 2H). Notably, HH hyperactivation induced by *Ptch1* depletion is abrogated by the treatment with Vismodegib (Fig 2I), an FDA-approved SMO antagonist that is used to treat HH-dependent tumors [12]. As expected, the chemical blockage of the HH cascade prevents SCO ventralization (Fig 2I).

In conclusion, we show a uniform differentiation of NPCs within SCOs by qPCR and IF. Marker expression and patterning of these NPCs are dependent on HH stimuli or genetic alterations and resemble the *in vivo* neural tube specification.

## Discussion

Spinal cord organoids (SCOs) have already been applied to study neural tube development [13], the differentiation of neural progenitors along the rostro-caudal axis [14] or to study late-stage neural specification [15]. In this paper, we focus on how SCOs can be applied to investigate the impact of genetic and chemical interventions on the HH signaling cascade. Applying an optimized protocol, based on the original report by Wichterle et al. [8], we show that SCOs recapitulate responses to genetic and chemical manipulations in the HH signaling in a quantifiable manner, thus making a comparison between different mutations and calibration possible. The *in vitro* assay involves the specification of NPCs along the dorsal-ventral axis and can be utilized to measure the effects of gene candidates on the HH pathway in a developmental-relevant context.

The size of nEBs and SCOs is of great importance for the expression of neural stemness markers such as *Sox1* and for establishing pattering in SCOs. To improve the sample preparation process, we use AggreWell plates which ensure the formation of nEBs of uniform size within the first three days of the experiment. This facilitates a high level of homogeneity between samples and biological replicates. These nEBs respond equally to RA treatment and give rise to SCOs with a consistent SOX1 expression. This improvement enables a focused analysis of HH regulation, reducing variability in neuroectoderm specification.

To characterize SCOs, we use a large panel of marker genes in an RT-qPCR approach to obtain a rapid and comprehensive overview of the expression pattern along the dorsal to the ventral axis of neural progenitor markers within our SCOs. An immunofluorescence-based characterization is then performed to evaluate protein levels of selected markers for different positions along the dorsal-ventral axis. The quantification of the NPC fractions expressing dorsal or ventral markers allows to categorize and quantify the HH response. A fraction of progenitors expresses the dorsal markers PAX7 or PAX6 and represents a level of HH signaling activity that would correspond to a first threshold of HH signaling activation, where GliR represses HH target gene expression. A second level threshold of increased HH signaling is observed through the expression of the first HH-dependent genes, such as *Olig2*. Strong and

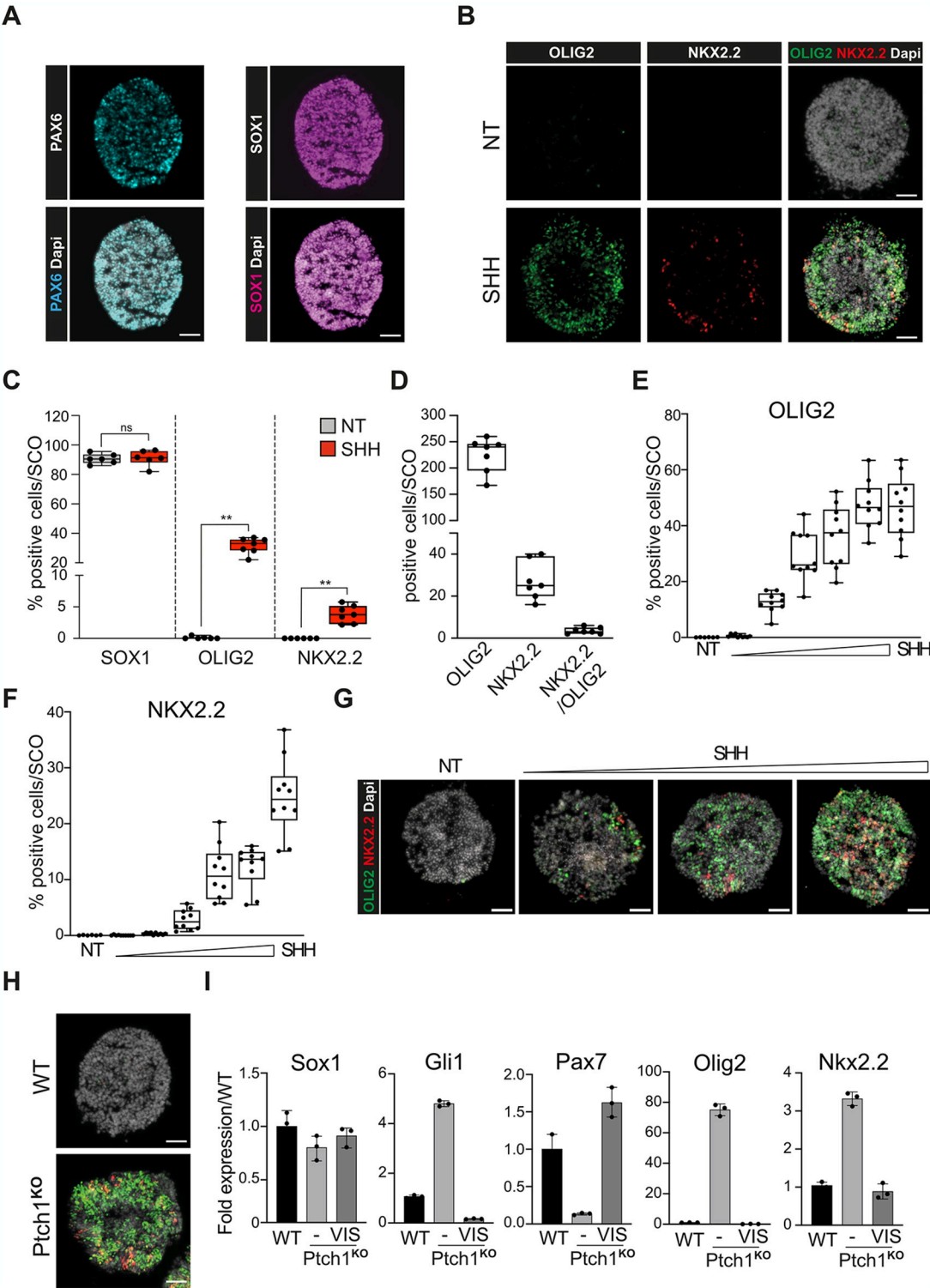

**Fig 2. Characterization of SCOs by immunostaining. A)** Expression of PAX6 (left) and SOX1 (right) markers in unstimulated SCO at day 6 of differentiation. The same SCO is shown for both staining. In the bottom panels, the overlay between PAX6 and SOX1 with nuclei (labeled with Dapi) is shown. Scale bar 50μm. **B)** SCOs express ventral markers after SHH stimulation. Expression of the ventral markers OLIG2 and NKX2.2 in SCOs treated with and without SHH. Nuclei are labeled with Dapi. Scale bar 50 μm. **C)** Quantification of NPCs expressing SOX1, OLIG2 and NKX2.2 markers in SCOs treated with and without SHH. Each point represents the percentage of positive cells in an independent SCO, relative to the total number of cells (mean +/-

SD) (n = 10). P value (**) < 0.01 (Welsch T-test). **D)** Quantification of NPCs co-expressing OLIG2 and NKX2.2 markers. Each point corresponds to the number of positive cells per SCO (mean ± SD) (n = 10). **E-F)** Quantification of NPCs expressing OLIG2 **(E)** and NKX2.2 **(F)** markers in SCOs treated with a range of SHH (10 ng/ml to 2 µg/ml) for 24 hours. Each point represents the percentage of positive cells in an independent SCO, relative to the total number of cells (mean ± SD) (n = 10). **G)** Representative pictures of SCOs quantified in E-F. Scale bar 50 µm. **H)** SCOs with a *Ptch1* mutation express ventral markers. Expression of the ventral markers OLIG2 and NKX2.2 in SCOs derived from WT and *Ptch1*^KO ESCs. Nuclei are labeled with Dapi. Scale bar 50 µm. **I)** SMO inhibition by chemicals prevents HH hyperactivation in *Ptch1*-depleted cells. Analysis by RT-qPCR of *Gli1*, the dorsal marker *Pax7* and the ventral marker *Olig2* in WT and *Ptch1*^KO SCOs treated for 48 hours with SHH or Vismodegib (1 µM) for 48 hours as indicated. Fold induction is relative to the WT sample (mean ± SD). *Eif4a2* expression was used for normalization. Each dot represents a biological replicate.

prolonged pathway activation then leads to the expression of the most ventral genes, such as *Nkx2.2* and *Foxa1* where the level of HH signal strength is above a third threshold. The effects of perturbations in HH signaling can be classified accordingly to these thresholds and compared between experiments.

Our SCOs show a similar marker expression response to SHH stimuli as the neural tube. SCOs initially have a dorsal identity but are responsive to ventralizing stimuli like SHH, which induces the expression of ventral markers and promotes the repression of dorsal ones (Fig 2B–2D). Mutations that constitutively activate the HH pathway, such as a *Ptch1* depletion (Fig 2G), promote a strong ventralization of SCOs, with more than 75% of NPCs expressing ventral markers. These results mirror defects detected in the neural tubes of *Ptch1*^KO embryos [16]. Further treatment with the SMO antagonist Vismodegib abrogates the HH hyperactivation in *Ptch1*^KO SCOs, showing that the SCO culture yields physiologically consistent and measurable responses.

Since ESCs can be easily modified by gene editing, SCOs with specific mutations can be generated to study the impact on the HH pathway. We have used SCOs in an earlier study that screened haploid ESCs by insertional mutagenesis [17] and discovered new modulators of the HH cascade [9]. Among them, we characterized the role of the COP-I receptor *Tmed2* as a negative modulator of the HH pathway [9]. The *Tmed2* mutation results in embryonic lethality, and it is impossible to study its effects in mouse models [9]. Thereby, *Tmed2* mutant SCOs enabled us to quantify the effect on HH signaling in neural patterning, revealing a ventralizing effect. In this case, two main advantages of this technique can be appreciated. First, SCOs can be used to study genes that cause major developmental defects that would make quantification impossible *in vivo*. Second, fast gene editing in ESCs allows a rapid analysis without the need for animal experiments.

An alternative to the here described SCOs is a 6-day protocol in 2D obtaining spinal cord precursors [18]. This technique has been used to characterize the role of novel candidates in regulating the HH pathway [19]. In this approach, the maturation of NPCs and their specification is highly dependent on the confluency within the samples, making the analysis and quantification of markers in our hands less robust and comparisons difficult. SCOs are largely unaffected by this issue as a homogenous confluency and marker expression throughout the whole sample and between replicates are observed. Therefore, SCOs are superior for quantitative analyses.

Fibroblast cells have been extensively used to investigate HH signaling. Their handling and the absence of requirements in stem cell knowledge make them optimal for a first insight into the role of a gene within the HH cascade. However, this system has limited use in studying the relevance of a gene in more physiological conditions. For example, while we observed a substantial upregulation of HH target genes in SCOs upon *Tmed2* depletion, we only detected a marginal increase of these genes in NIH-3T3 cells [9]. Combining several in vitro culture systems is preferred to substantiate mechanistic insights for new HH signaling modulators. Overall, we suggest SCOs as a quantitative and sensitive method to study regulators of the HH

pathway in a relevant mammalian developmental system. Quantitation of ventral and dorsal markers recapitulates the spectrum of cell types during neural tube specification and offers a reliable and robust method to quantify the HH signaling.

## Supporting information

**S1 File.**
(PDF)

## Acknowledgments

We thank all the members of the Wutz lab for help with reagents and discussions.

## Author Contributions

**Conceptualization:** Markus Holzner, Anton Wutz, Giulio Di Minin.

**Data curation:** Markus Holzner, Giulio Di Minin.

**Formal analysis:** Markus Holzner, Giulio Di Minin.

**Funding acquisition:** Anton Wutz, Giulio Di Minin.

**Investigation:** Markus Holzner, Giulio Di Minin.

**Methodology:** Markus Holzner, Anton Wutz, Giulio Di Minin.

**Project administration:** Anton Wutz, Giulio Di Minin.

**Resources:** Markus Holzner, Anton Wutz, Giulio Di Minin.

**Supervision:** Anton Wutz, Giulio Di Minin.

**Validation:** Markus Holzner, Giulio Di Minin.

**Visualization:** Markus Holzner, Giulio Di Minin.

**Writing – original draft:** Markus Holzner, Anton Wutz, Giulio Di Minin.

**Writing – review & editing:** Markus Holzner, Anton Wutz, Giulio Di Minin.

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
