## [Decision Letter · Decision Letter 0]

23 Oct 2023

PONE-D-23-20299Applying Spinal Cord Organoids as a quantitative approach to study the mammalian Hedgehog pathwayPLOS ONE

Dear Dr. Di Minin,

Thank you for submitting your manuscript to PLOS ONE. After careful consideration, we feel that it has merit but does not fully meet PLOS ONE’s publication criteria as it currently stands. Therefore, we invite you to submit a revised version of the manuscript that comprehensively addresses the points raised during the review process. As both reviewers have voiced serious concerns about the manuscript in its current form, the revised manuscript needs to address the reviewers' comments in full in order to warrant publication in PLOS ONE.

We look forward to receiving your revised manuscript.

Kind regards,

Michael Schubert

Academic Editor

PLOS ONE

2. We note that Figure 1B in your submission contain copyrighted images. All PLOS content is published under the Creative Commons Attribution License (CC BY 4.0), which means that the manuscript, images, and Supporting Information files will be freely available online, and any third party is permitted to access, download, copy, distribute, and use these materials in any way, even commercially, with proper attribution. For more information, see our copyright guidelines: http://journals.plos.org/plosone/s/licenses-and-copyright.

1. You may seek permission from the original copyright holder of Figure 1B to publish the content specifically under the CC BY 4.0 license.

Reviewers' comments:

Reviewer's Responses to Questions

**Comments to the Author**

1. Does the manuscript report a protocol which is of utility to the research community and adds value to the published literature?

Reviewer #1: No

Reviewer #2: No

2. Has the protocol been described in sufficient detail?

To answer this question, please click the link to protocols.io in the Materials and Methods section of the manuscript (if a link has been provided) or consult the step-by-step protocol in the Supporting Information files.

The step-by-step protocol should contain sufficient detail for another researcher to be able to reproduce all experiments and analyses.

Reviewer #1: Yes

Reviewer #2: Yes

3. Does the protocol describe a validated method?

Reviewer #1: Yes

Reviewer #2: Yes

4. If the manuscript contains new data, have the authors made this data fully available?

Reviewer #1: Yes

Reviewer #2: Yes

**5. Is the article presented in an intelligible fashion and written in standard English?**

Reviewer #1: Yes

Reviewer #2: Yes

6. Review Comments to the Author

Reviewer #1: Unfortunately, the added value of this protocol to the spinal cord organoid scientific community is very limited, in my view. The most novel point is probably the use of Aggrewell plates, which have been used in numerous organoid-based studies but not for SCOs. Apart from that, how the HH pathway determines the dorso-ventral patterning of the developing spinal cord, using 2D and 3D models, have been shown numerous times before, although probably not as a sole-focus in a compiled Methods paper like this one. To increase the value of the study, a more thorough study using a range of SHH and dorsal morphogens concentrations could be performed.

The text would need some general editing to improve its flow. For example already in the abstract, saying that “The Hedgehog pathway… is implicated as the underlying cause of numerous diseases” sounds weird. The authors mean that alterations in the pathway have been reported as underlying cause of XYZ. / Ptch1 is mentioned but not described, and saying that SCOs are a “physiological context” is a bit far-reaching because it’s not as physiological as in vivo. Better to say in comparison with commonly used, more simplistic 2D cellular models.

Other examples, just in the introduction:

- Line 58: this sentence is confusing: “Additionally, certain anatomical features that are regulated by the HH gene are only found in invertebrates and vertebrates”. I guess the authors mean “found in either or”?

- Line 62: where does “here” refer to?, the “vertebrae development”?, here as “in this study”?

- Line 63: why SHH instead of HH?, the authors should explain.

- Line 87: “SCO” not described, not explained from which type/species of pluripotent stem cells they are derived.

- Line 90: “SCOs… exhibit equal properties in the dorsal to ventral patterning as neural tubes do.” It’s unclear what the authors refer to by “equal properties”.

- Line 92: “…novel capacities of TMED2 within the HH signaling cascade”. TMED2 is not described.

In the result sections/discussion:

- Apparently the SCOs are embedded (line 140), but to that point it’s not described how/in what, or do the authors mean SCO cryosections and therefore, it’s inferred that the SCOs are embedded in OCT (?) for sectioning? Unclear.

- Line 150, what do they authors mean by “To study mechanistic influences within the HH pathway”? That is also poorly written. And, are the “Ptch1KO ESCs” mouse or human?

- Line 179, the authors mention different SCO formation efficiency in different “cellular backgrounds” but don’t explain which ones are those, nor there is no reference to that in the results.

- The discussion would need quite some work too. It starts by repeating what said in the results section. In addition, an entire paragraph is devoted to describe a previous study by the group on Tmed2, which seems very out of context. Further, numerous studies have used similar protocols (including EBs/organoids) to model spinal cord dorso-caudal domain specification. These papers should cited and how this study provides additional knowledge on this particular topic discussed. For example:

Amin 2023 - doi: 10.1101/2023.05.31.541819 bioRxiv 2023

Mouilleau 2021 - doi:10.1242/dev.194514

Ogura 2018 - doi:10.1242/dev.162214

Finally, the last statement is also confusing, “While the dorsal-ventral axis is not recreated in the in vitro settings…”. I understood that the main claim of the study is that an organoid model can be used to investigate the role of SHH signaling pathway in the dorso-ventral patterning of the spinal cord development. That statement contradicts the later claim.

Figures:

- Figure 1E and entire Figure 2: what day of the differentiation protocol are those SCOs?

- Housekeeping gene/s used for qPCR analysis not mentioned

- What does “fold expression” mean? / if the values are relative to one of the conditions (i.e. “WT”), why is that conditions not at 1.0 in all cases?

Regarding the protocol itself, it should be as detailed as possible, therefore indications like “…plate the desired amount in ESC media” are not appropriate.

Also, for an sterile, long-term culture like organoids, I doubt that “bacterial plates to prevent nEBs from attaching to the plate” is the best approach. These cultures are routinely done in ultra-low attachment dishes (all dimensions available from multiple companies nowadays).

Reviewer #2: Markus Holzner and colleagues report an optimized protocol for generating spinal cord organoids (SCOs). Overall, the current manuscript suffers from being a relatively straight forward replication of Wichterle et al. (https://www.sciencedirect.com/science/article/pii/S0092867402008358). The SCOs are obtained in almost the same fashion as embryoid bodies by Wichterle et al, and, indeed, the material and methods section of this manuscript even refers to them as embryoid bodies. Figure 1 uses qRT-PCR to demonstrate that the SCOs are responsive to SHH, and most of Figure 2 uses immunofluorescence to show the same thing. Together, this is essentially a repeat of Figure 1 of Wichterle et al. Figures 2E and 2F include the use of Ptch1 knockout cells, but this isn’t sufficiently novel since Ptch1 is well-known to be the SHH receptor. The manuscript claims to have optimized the SCO procedure, but it would help to see a description of how the protocol has been improved over Wichterle et al.

I also wonder if the current manuscript is somehow incomplete. The discussion section refers to using SCOs to demonstrate a strong ventralizing effect of Tmed2 knockout (line 222). However, I cannot find this in the manuscript – neither in the results section, nor in the figures. Although the authors cited their previous work on Tmed2 (Minn et al.), that study seems to use 2D cultures and not 3D cultures. Perhaps the authors intended to include more data in this manuscript for Tmed2?

Since the authors are claiming that SCOs can be used as a model system for studying the Hh pathway, it would help to see more data on Hh pathway components. For example, cilium formation by cells in SCOs as well as the presence of activator and repressor Gli forms, including at different Hh concentrations. This would better establish the tool set for studying the Hh pathway in this system.

Additional minor comments are:

• Line 39: “miss regulations” should read, “misregulation.”

• Should include citations for line 41 for the claim “frequently observed”.

• There is a strange formatting regarding carriage returns. For example, there is no carriage return between lines 42 and 43, whereas there is a carriage return for line 51. Along similar lines, it looks like some of these may interrupt paragraphs. For example, lines 115-117 is a single sentence. Was this really intended as a full paragraph? This happens repeatedly throughout the text (another example being between lines 140 and 141).

• Sentence in lines 54 – 57 needs a citation (starts, “Validation of these candidates in mammals”)-

• Sentence in line 57 needs a citation, “Key structures like the primary cilium are unique to vertebrates”.

• Line 63: SHH is also secreted from the floor plate of the neural tube.

• Line 67: “GLI code” was coined by Altaba et al., and their paper should be cited here. https://www.ncbi.nlm.nih.gov/pmc/articles/PMC2601665/

• Lines 67-68. Would be better to describe the generation of GliR forms in more detail, e.g., by cleavage of the full length form, which has activation activity.

• Line 69: seems to be a misunderstanding. There are many more than just three different neural progenitor fates along the dorsoventral axis that are specified by SHH signaling. The cited paper refers to the SHH gradient being “incremental two- to threefold change in Shh concentration”. Maybe this is what was meant?

• Maybe I missed it, but which mouse ESC line was used?

• Images in 2A make the DAPI hard to see. Would be better to see each panel separately and then merge.

• Figure legends are missing the description of the tests used for statistical significance as well as what the error bars indicate and what the stars indicate. Figure legends should also indicate how many biological replicates were performed.

7. PLOS authors have the option to publish the peer review history of their article (what does this mean?). If published, this will include your full peer review and any attached files.

Reviewer #1: No

Reviewer #2: No

---

## [Author Response · Author response to Decision Letter 0]

8 Jan 2024

"Applying Spinal Cord Organoids as a quantitative approach to study the mammalian Hedgehog pathway" by Holzner et al. (PONE-D-23-20299).

Response to the editorial comments

After careful consideration, we feel that it has merit but does not fully meet PLOS ONE’s publication criteria as it currently stands. Therefore, we invite you to submit a revised version of the manuscript that comprehensively addresses the points raised during the review process. As both reviewers have voiced serious concerns about the manuscript in its current form, the revised manuscript needs to address the reviewers' comments in full in order to warrant publication in PLOS ONE.

Response: We would like to thank the reviewers and the editor for their comments, which have helped us to further strengthen our manuscript. In particular, we have performed two additional experiments to address the reviewers’ requests. These experiments are shown in new panels in Figure 1D and Figure 2D-E. In addition, we have made changes to the text to clarify and focus our manuscript. We responded to all points raised by the reviewers below:

Response to Reviewer #1:

Unfortunately, the added value of this protocol to the spinal cord organoid scientific community is very limited, in my view. The most novel point is probably the use of Aggrewell plates, which have been used in numerous organoid-based studies but not for SCOs. Apart from that, how the HH pathway determines the dorso-ventral patterning of the developing spinal cord, using 2D and 3D models, have been shown numerous times before, although probably not as a sole-focus in a compiled Methods paper like this one. 

Response: We acknowledge that multiple papers are available describing SCO derivation and the role of morphogens (SHH or BMP) in dorsal-ventral patterning. We believe that our method has a specific value for the scientific community investigating the mechanisms of HH signaling regulation. At the moment, the functional characterization of genes linked to HH signaling is performed in fibroblast/cancer cells or in mouse models (limb and neural tube development). As specified in the manuscript, both approaches have strengths and limitations. We propose SCO application as a suitable system for quantifying HH signaling in a developmental context. Importantly, this does not require extensive time (measurements can be performed in a week) and animal experiments. We now show that the quantification of HH signaling intensity can be calibrated. Providing a robust and detailed method on how to derive SCOs and characterize HH pathway activation, we support its application in the field. Following the reviewer’s suggestion, in the new version of the manuscript, we further acknowledge the existence of additional works describing SCO derivation and stress the value of this method paper for the HH field (1st paragraph of the Discussion). Considering the ease of introducing genetic changes in ESCs, we anticipate the adoption of our strategy in the community.

To increase the value of the study, a more thorough study using a range of SHH and dorsal morphogens concentrations could be performed. 

Response: We thank the reviewer for this suggestion. We have now performed a new experiment (Fig 2E-G) using a range of SHH and characterized the differential expression of the ventral markers Olig2 and Nkx2.2 in SCOs. Treatment with dorsal morphogens is certainly important to better characterize the dorsal patterning of SCOs. However, at present, we feel that this would likely distract from our focus on the HH pathway. We are planning to perform an extensive study in the future using dorsal signals.

The text would need some general editing to improve its flow. For example already in the abstract, saying that “The Hedgehog pathway… is implicated as the underlying cause of numerous diseases” sounds weird. The authors mean that alterations in the pathway have been reported as underlying cause of XYZ. / Ptch1 is mentioned but not described, and saying that SCOs are a “physiological context” is a bit far-reaching because it’s not as physiological as in vivo. Better to say in comparison with commonly used, more simplistic 2D cellular models. 

Response: We agree that our phrasing was misleading and have now modified the text following the reviewer's suggestion (Abstract)

Other examples, just in the introduction:

- Line 58: this sentence is confusing: “Additionally, certain anatomical features that are regulated by the HH gene are only found in invertebrates and vertebrates”. I guess the authors mean “found in either or”? 

Response: We modified the text according to the reviewer’s comment (3rd paragraph of the Introduction).

- Line 62: where does “here” refer to?, the “vertebrae development”?, here as “in this study”? Response: This point was not clear enough, and according to the reviewer’s concerns, we have now modified the phrase in the manuscript (3rd paragraph of the Introduction)

- Line 63: why SHH instead of HH?, the authors should explain. 

Response: We apologize for the lack of clarity. A sentence was added to explain the existence of three HH-secreted molecules in mammalian cells (2nd paragraph of the Introduction)

- Line 87: “SCO” not described, not explained from which type/species of pluripotent stem cells they are derived. 

Response: This information is mentioned in the new version of the manuscript (1st paragraph of the Results)

- Line 90: “SCOs… exhibit equal properties in the dorsal to ventral patterning as neural tubes do.” It’s unclear what the authors refer to by “equal properties”. 

Response: We substituted “equal properties” with “lead to the specification of neural progenitors with dorsal and ventral cellular identities, like in the neural tube.” (end of the 5th paragraph of the Introduction).

- Line 92: “…novel capacities of TMED2 within the HH signaling cascade”. TMED2 is not described. 

Response: In the new version, a sentence introduces TMED2 in the context of the HH signaling (last paragraph of the Introduction).

In the result sections/discussion:

- Apparently the SCOs are embedded (line 140), but to that point it’s not described how/in what, or do the authors mean SCO cryosections and therefore, it’s inferred that the SCOs are embedded in OCT (?) for sectioning? Unclear. 

Response: We thank the reviewer for pointing this out. SCOs are embedded in OCT and sectioned for analysis. To avoid adding technical details in the result description, we excluded the confusing “embedded” information from the new version of the manuscript (3rd paragraph of the Results). The necessary details are already available in the method/protocol section.

- Line 150, what do they authors mean by “To study mechanistic influences within the HH pathway”? That is also poorly written. 

Response: The phrase was modified to improve clarity (4th paragraph of the Results). 

And, are the “Ptch1KO ESCs” mouse or human?

Response: The missing information was added to the text (4th paragraph of the Results).

- Line 179, the authors mention different SCO formation efficiency in different “cellular backgrounds” but don’t explain which ones are those, nor there is no reference to that in the results. 

Response: We thank the reviewer for pointing out the missing information. In the new version, the statement was removed (2nd paragraph of the Discussion).

- The discussion would need quite some work too. It starts by repeating what said in the results section. In addition, an entire paragraph is devoted to describe a previous study by the group on Tmed2, which seems very out of context. Further, numerous studies have used similar protocols (including EBs/organoids) to model spinal cord dorso-caudal domain specification. These papers should cited and how this study provides additional knowledge on this particular topic discussed. For example: 

Amin 2023 - doi: 10.1101/2023.05.31.541819 bioRxiv 2023 

Mouilleau 2021 - doi:10.1242/dev.194514 

Ogura 2018 - doi:10.1242/dev.162214

Response: We agree with the reviewer that our discussion can be strengthened by including the additional studies. To address the reviewer’s comment, we made the following changes to the text: 1) we summarized the paragraph describing the role of Tmed2 in HH signaling. According to the journal guidelines, the methodology should be supported by novel experiments or/and citing a work in which the approach has been applied. In the result section, we described novel experiments supporting the methodology, and in the discussion, we additionally describe our recent work where we used SCOs to characterize a novel modulator of the HH cascade. We introduced Tmed2 to provide a clear example of the benefits of using SCOs to study HH signaling modulators. 2) Following in the discussion, we additionally compare the SCO model to established fibroblast and 2D cellular systems used to investigate the HH pathway in mammalian cells. We explain how this study provides additional knowledge on this specific topic discussed (i.e. How to investigate HH signaling in mammalian cells). 3) We thank the reviewer for pointing out the need to reference other works that describe protocols to model spinal cord dorso-caudal domain specification. The references were added in the new version of the manuscript (1st paragraph of the Discussion). 

Finally, the last statement is also confusing, “While the dorsal-ventral axis is not recreated in the in vitro settings…”. I understood that the main claim of the study is that an organoid model can be used to investigate the role of SHH signaling pathway in the dorsoventral patterning of the spinal cord development. That statement contradicts the later claim. 

Response: We thank the reviewer for this rightful comment. In the new version, the statement was modified to improve clarity (last paragraph of the Discussion).

Figures:

- Figure 1E and entire Figure 2: what day of the differentiation protocol are those SCOs?

Response: In the revised figure legends, the day of differentiation is now clearly indicated.

- Housekeeping gene/s used for qPCR analysis not mentioned. 

Response: The housekeeping genes used for qPCR are indicated in the method section. For improving clarity, we also indicated them in the figure legends.

- What does “fold expression” mean? / if the values are relative to one of the conditions (i.e. “WT”), why is that conditions not at 1.0 in all cases? 

Response: Data were normalized to one replicate of the experiment. In the new figure version, we use the average of WT samples for normalization. 

Regarding the protocol itself, it should be as detailed as possible, therefore indications like “…plate the desired amount in ESC media” are not appropriate. 

Response: We now provide detailed information in the revised version.

Also, for an sterile, long-term culture like organoids, I doubt that “bacterial plates to prevent nEBs from attaching to the plate” is the best approach. These cultures are routinely done in ultra-low attachment dishes (all dimensions available from multiple companies nowadays). 

Response: We made the statement more generic to accommodate different culture conditions used in laboratories. 

Response to Reviewer #2

Markus Holzner and colleagues report an optimized protocol for generating spinal cord organoids (SCOs). Overall, the current manuscript suffers from being a relatively straight forward replication of Wichterle et al. (https://www.sciencedirect.com/science/article/pii/S0092867402008358). The SCOs are obtained in almost the same fashion as embryoid bodies by Wichterle et al, and, indeed, the material and methods section of this manuscript even refers to them as embryoid bodies. Figure 1 uses qRT-PCR to demonstrate that the SCOs are responsive to SHH, and most of Figure 2 uses immunofluorescence to show the same thing. 

Response: Our study evaluates SCOs for quantifying the effect of mutations and interventions on mammalian Hedgehog pathway”. Although, SCOs have been established previously we believe that our report is important for the scientific community investigating the mechanism of HH signaling regulation. Presently, fibroblast/cancer cells or mouse models (limb and neural tube development) are used to investiagate HH signaling. SCOs provide advantages of a developmental context and allow rapid measurement (in one week) without a need for animal experiments. In addition, quantification is straight forward in sharp contrast to using developmental systems. In our revised version we now demonstrate that quantification can be calibrated. Together with the ease of engineering genetic modifications in ESCs we anticipate adoption of SCOs for studying HH signaling in the wider scientific community. Providing a robust and detailed method on how to derive SCOs and characterize HH pathway activation, we support its application in the field. Following the reviewer’s point, in the new version of the manuscript, we stress the value of this method paper for the HH field (1st and 4th paragraph of the Discussion). As pointed out by the reviewer, in this manuscript, we show two different strategies to quantify the intensity of HH response: by IF and RT-qPCR. We provide a detailed description of how to optimize EB formation and sectioning to obtain quantifiable pictures by IF. These details are not present in Wichterle et al but are crucial for the users who intend to use SCOs to quantify HH signaling regulation. 

Together, this is essentially a repeat of Figure 1 of Wichterle et al. Figures 2E and 2F include the use of Ptch1 knockout cells, but this isn’t sufficiently novel since Ptch1 is well-known to be the SHH receptor. 

Response: To prove the validity of SCOs as a tool to quantify HH signaling response, we applied genetic (Ptch1 depletion) and chemical triggers well-known to activate or repress the HH signaling cascade. These positive controls are used to demonstrate the robustness of the method for investigating perturbations in the intensity of HH response. 

The manuscript claims to have optimized the SCO procedure, but it would help to see a description of how the protocol has been improved over Wichterle et al. 

Response: In Figure 1 of Wichterle et al. mentioned before by the reviewer, the neuralized EBs (or SCOs) don’t show a homogenous SOX1 expression. Neuralization seems to occur mainly at the edges of the EBs. This issue complicates the use of SCOs to quantify HH signaling as only a subset of neural progenitors may respond to SHH. SCOs of homogenous size obtained using Aggrewell plates show more than 90% of SOX1 expression (Fig 2A). This improvement is crucial to allow quantitative analysis of the HH response. We made clear this point in the new manuscript version (2nd paragraph of the Discussion).

I also wonder if the current manuscript is somehow incomplete. The discussion section refers to using SCOs to demonstrate a strong ventralizing effect of Tmed2 knockout (line 222). However, I cannot find this in the manuscript – neither in the results section, nor in the figures. Although the authors cited their previous work on Tmed2 (Minn et al.), that study seems to use 2D cultures and not 3D cultures. Perhaps the authors intended to include more data in this manuscript for Tmed2?

Response: According to the journal guidelines, the methodology should be supported by novel experiments or/and citing a work in which the approach has been applied. In the result section, we described novel experiments supporting the methodology, and in the discussion, we additionally reviewed our recent work where we used SCOs to characterize a novel modulator of the HH cascade. We introduced Tmed2 to provide a clear example of the benefits of using this model to study HH signaling modulators. In Di Minin et al., 2022 (doi:10.1371/journal.pbio.3001596), the SCO methodology was applied to characterize the role of Tmed2 as a negative regulator of the HH cascade. In the panels E and F of Figure 4 of Di Minin et al., 2022, the results presented were obtained using SCOs (called in the early version as neuralized EBs) obtained from WT, Smo-/- and Tmed2-/- ESCs. This is stated in the figure legend and in the result section (Chapter: “TMED2 is a negative regulator of HH signaling in neural differentiation”). In the cha

---

## [Decision Letter · Decision Letter 1]

16 Feb 2024

PONE-D-23-20299R1Applying Spinal Cord Organoids as a quantitative approach to study the mammalian Hedgehog pathwayPLOS ONE

Dear Dr. Di Minin,

Thank you for submitting your manuscript to PLOS ONE. After careful consideration, we feel that it has merit but does not fully meet PLOS ONE’s publication criteria as it currently stands. Therefore, we invite you to submit a revised version of the manuscript that addresses the points raised during the review process.

We look forward to receiving your revised manuscript.

Kind regards,

Michael Schubert

Academic Editor

PLOS ONE

Journal Requirements:

Reviewers' comments:

Reviewer's Responses to Questions

**Comments to the Author**

1. Does the manuscript report a protocol which is of utility to the research community and adds value to the published literature?

Reviewer #1: Yes

Reviewer #2: Yes

2. Has the protocol been described in sufficient detail?

To answer this question, please click the link to protocols.io in the Materials and Methods section of the manuscript (if a link has been provided) or consult the step-by-step protocol in the Supporting Information files.

The step-by-step protocol should contain sufficient detail for another researcher to be able to reproduce all experiments and analyses.

Reviewer #1: Yes

Reviewer #2: Yes

3. Does the protocol describe a validated method?

Reviewer #1: Yes

Reviewer #2: Yes

4. If the manuscript contains new data, have the authors made this data fully available?

Reviewer #1: Yes

Reviewer #2: Yes

**5. Is the article presented in an intelligible fashion and written in standard English?**

Reviewer #1: Yes

Reviewer #2: Yes

6. Review Comments to the Author

Reviewer #1: The authors have addressed all concerns previously raised and significantly improved the quality of the study. I just have 2 minor points that should be easily tackled without the need of further experiments. Please, see below.

Unless I'm reading this wrong, the claim in lines 210-211 ("Additionally, NKX2.2 represses the Olig2 gene, which is

coherent with the in vivo situation") is not supported by the results. Actually Fig 2E-F shows a continuous increase OLIG2 AND NKX2.2 expression (if at the highest [SHH] NKX2.2 was highest and OLIG2 was lower than a lower [SHH] one could make that extrapolation (not a claim and just based on in vivo knowledge), but actually OLIG2 never decreases in that plot.

So I would just indicate that the expression of both ventral progenitor markers increases according to increasing [SHH], without entering in OLIG2 inhibition by NKX2.2 (maybe even higher [SHH] would have needed to be tested, or the expression of the markers checked a few days later).

Also, it's a pity that the images in Fig 1D poorly show the primary cilia, especially upon SHH treatment. Ideally, better images should be shown, if possible.

Reviewer #2: The authors have adequately addressed the requested points. The manuscript has been modified to better describe the cell lines used, the background material, as well as better highlight the use of AggreWell in their protocol compared to the existing literature.

7. PLOS authors have the option to publish the peer review history of their article (what does this mean?). If published, this will include your full peer review and any attached files.

Reviewer #1: No

Reviewer #2: No

---

## [Author Response · Author response to Decision Letter 1]

19 Mar 2024

Response to the editorial comments

Thank you for submitting your manuscript to PLOS ONE. After careful consideration, we feel that it has merit but does not fully meet PLOS ONE’s publication criteria as it currently stands. Therefore, we invite you to submit a revised version of the manuscript that addresses the points raised during the review process.

Again, we would like to thank the reviewers and the editor for their comments, which have helped us strengthen our manuscript further. We responded to all the individual points raised by Reviewer #1. As suggested, we modified the images shown in Fig1D to better show SMO translocation at the PC upon SHH treatment.

Response to Reviewer #1

The authors have addressed all concerns previously raised and significantly improved the quality of the study.

Response: We thank the reviewer for his positive assessment of our revision.

I just have 2 minor points that should be easily tackled without the need of further experiments. Please, see below.

Unless I'm reading this wrong, the claim in lines 210-211 ("Additionally, NKX2.2 represses the Olig2 gene, which iscoherent with the in vivo situation") is not supported by the results. Actually Fig 2E-F shows a continuous increase OLIG2 AND NKX2.2 expression (if at the highest [SHH] NKX2.2 was highest and OLIG2 was lower than a lower [SHH] one could make that extrapolation (not a claim and just based on in vivo knowledge), but actually OLIG2 never decreases in that plot. So I would just indicate that the expression of both ventral progenitor markers increases according to increasing [SHH], without entering in OLIG2 inhibition by NKX2.2 (maybe even higher [SHH] would have needed to be tested, or the expression of the markers checked a few days later).

Response: We have followed the reviewer’s comment and modified the phrasing as suggested (line 210). 

Also, it's a pity that the images in Fig 1D poorly show the primary cilia, especially upon SHH treatment. Ideally, better images should be shown, if possible.

Response: We provide new images (Fig1D) that better support SMO translocation upon SHH treatment. Compared to experiments performed in fibroblast cells, SCO sections are not optimal for characterizing SMO translocation at the primary cilium due to two drawbacks of the methodology: 1) the majority of NPCs in SCOs are in a proliferative stage, which means that either they do not have any PCs or the PCs are not long enough to be distinguished clearly (as they are more associated with the G0 phase). 2) the sections that are produced during the slicing of the EBs are usually around 10-15μm thick, which may not include the necessary planes to detect the PCs entirely.

Response to Reviewer #2

The authors have adequately addressed the requested points. The manuscript has been modified to better describe the cell lines used and the background material, as well as better highlight the use of AggreWell in their protocol compared to the existing literature.

Response: We thank the reviewer for the specific comments and the positive evaluation of our study.

---

## [Editor Report · Decision Letter 2]

20 Mar 2024

Applying Spinal Cord Organoids as a quantitative approach to study the mammalian Hedgehog pathway

PONE-D-23-20299R2

Dear Dr. Di Minin,

We’re pleased to inform you that your manuscript has been judged scientifically suitable for publication and will be formally accepted for publication once it meets all outstanding technical requirements.

Kind regards,

Michael Schubert

Academic Editor

PLOS ONE

---

## [Editor Report · Acceptance letter]

13 Jun 2024

PONE-D-23-20299R2 

PLOS ONE

Dear Dr. Di Minin, 

I'm pleased to inform you that your manuscript has been deemed suitable for publication in PLOS ONE. Congratulations! Your manuscript is now being handed over to our production team.

Kind regards, 

on behalf of

Dr. Michael Schubert 

Academic Editor

PLOS ONE